Conservation gaps and priorities of range-restricted birds in the Northern Andes

Medina Wilderson wam27@duke.edu 1
Pimm Stuart L. 1
Huang Ryan M. 1 2
1 Nicholas School of the Environment, Duke University , Durham , NC , United States of America
2 Conservation Ecology Research Unit, Department of Zoology and Entomology, University of Pretoria , Hatfield , South Africa
Nazareno Alison
Electronic publication date: 2024 Feb 26
Publication date: 2024
Volume: 12
Electronic Location ID: e16893
Received 2023 May 25; Accepted 2024 Jan 16
Copyright: ©2024 Medina et al.
Copyright year: 2024
Copyright holder: Medina et al.
License: This is an open access article distributed under the terms of the Creative Commons Attribution License, which permits unrestricted use, distribution, reproduction and adaptation in any medium and for any purpose provided that it is properly attributed. For attribution, the original author(s), title, publication source (PeerJ) and either DOI or URL of the article must be cited.
License URL: https://creativecommons.org/licenses/by/4.0/

Keywords: Area of habitat, Community lands, Crowd-sourced data, Protected areas, Species distribution range, Species extinction, Endangered species

Funding: The Fulbright - Pasaporte a la Ciencia fellowship Wilderson Medina was funded by the Fulbright - Pasaporte a la Ciencia fellowship. The authors received no funding for this work. The funders had no role in study design, data collection and analysis, decision to publish, or preparation of the manuscript.

==============================
The ongoing destruction of habitats in the tropics accelerates the current rate of species extinction. Range-restricted species are exceptionally vulnerable, yet we have insufficient knowledge about their protection. Species’ current distributions, range sizes, and protection gaps are crucial to determining conservation priorities. Here, we identified priority range-restricted bird species and their conservation hotspots in the Northern Andes. We employed maps of the Area of Habitat (AOH), that better reflect their current distributions than existing maps. AOH provides unprecedented resolution and maps a species in the detail essential for practical conservation actions. We estimated protection within each species’ AOH and for the cumulative distribution of all 335 forest-dependent range-restricted birds across the Northern Andes. For the latter, we also calculated protection across the elevational gradient. We estimated how much additional protection community lands (Indigenous and Afro-Latin American lands) would contribute if they were conservation-focused. AOHs ranged from 8 to 141,000 km2. We identified four conservation priorities based on cumulative species richness: the number of AOHs stacked per unit area. These priorities are high-resolution mapped representations of Endemic Bird Areas for the Tropical Andes that we consider critically important. Protected areas cover only 31% of the cumulative AOH, but community lands could add 19% more protection. Sixty-two per cent of the 335 species have ranges smaller than their published estimates, yet IUCN designates only 23% of these as Threatened. We identified 50 species as top conservation priorities. Most of these concentrate in areas of low protection near community lands and at middle elevations where, on average, only 34% of the land is protected. We highlight the importance of collaborative efforts among stakeholders: governments should support private and community-based conservation practices to protect the region with the most range-restricted birds worldwide.

Introduction

Approximately 2,000 bird species live across the Northern Andes; almost 30% have small geographical ranges and are unique to the region (Herzog & Kattan, 2012; Pimm et al., 2014). The western Andean corridor between Ecuador and Colombia is the richest range-restricted bird region in the world (Wilson et al., 2022) and one of the most severely impacted Endemic Bird Areas (EBA) in recent decades (Correa Ayram et al., 2020; Fjeldså& Irestedt, 2009; Jenkins, Pimm & Joppa, 2013). Range-restricted species —including small-ranged and endemic species, some recently described or not yet discovered—are highly susceptible to extinction. They must be a priority for conservation (Jenkins et al., 2015; Liu et al., 2022).

Surprisingly, despite being susceptible to threats due to their limited distribution, the International Union for Conservation of Nature (IUCN) Red List does not currently categorise many as Threatened—the combined categories of Critically Endangered, Endangered, and Vulnerable (Ocampo-Peñuela et al., 2016). In the Western Andes of Colombia, for instance, Ocampo-Peñuela & Pimm (2014) found large range reductions of several of these species’ estimated range sizes after refining published maps by the species’ known elevation and habitat preferences. Such work highlights the importance of working with refined maps that accurately reflect their current geographical boundaries. The Area of Habitat (AOH)—the habitat and elevation within a species’ range—provides essential updated information on where a species is likely found. It uses curated field observations, crowd-sourced data, and the current extent of their habitats and altitudinal ranges (Brooks et al., 2019; Huang et al., 2021). We use AOHs to identify species with small habitat ranges within a pool of range-restricted species and compare them with published ranges. AOH provides unprecedented resolution essential for practical conservation actions.

Beyond range size, setting conservation priorities must also estimate the extent of protection within each species’ range (Jenkins et al., 2015). Recent studies prioritising bird conservation areas in the Neotropics point to the Western Andes transition between Colombia and Ecuador as one of the least protected areas (Ocampo-Peñuela & Pimm, 2014; Wilson et al., 2022). Such analyses require an accurate systematisation of protected areas across the region. The World Database of Protected Areas (WDPA) offers unrivalled information on protected areas, but it is incomplete. The most elusive information on protected areas is that from private reserves. Efforts to assemble this information are still in their early stages (Bingham et al., 2021). Indigenous and Afro-Latin American lands (henceforth “community lands” when combined) are included in the WDPA, knowing they are not strictly but sometimes de facto protected areas (Wilson et al., 2022). They could contribute to the protection of species, but their conservation status is still debatable. For instance, the recent peace agreement in Colombia has allowed loggers and illegal croppers to access Afro-Colombian territories, thus urging an increased state presence and improved legal actions for their protection (Prem, Saavedra & Vargas, 2020; Wilson et al., 2022).

In addition to varying governance, we were also interested in identifying the protection status across their elevational ranges. Given the inherent connection between elevation and warming, several studies have shown climate-induced range shifts in tropical bird species (Freeman et al., 2018; Neate-Clegg et al., 2021). We have previously found mainly middle- to high-elevation range-restricted species significantly retreating from their lower ranges in the Northern Andes (Medina, Huang & Pimm, 2023).

This study aims to identify range-restricted bird conservation priorities and gaps to inform management decisions across the Northern Andes. How extensive are these protection gaps, and what other regions across the Northern Andes urgently need protection? We answer these questions using the cumulative distributions of the species most at risk across state, private, and community lands. We offer a list of conservation priorities based on limited range size and protection, reveal the regions with higher concentrations of range-restricted species, and suggest management conservation actions.

Materials & Methods

The Northern Andes lie on the northernmost extreme of South America and make up a large portion of the Tropical Andes biodiversity hotspot from Venezuela and Panamá to north-western Perú, where most range-restricted species occur. It is well-known for its exceptional biodiversity but has high human population densities. Our study involves range-restricted bird species with published ranges partially or entirely contained by the Northern Andes region, as Griffith, Omernik & Azevedo (1998) define it. Some 335 forest-dependent bird species with published ranges <50,000 km2 are native there.

Mapping area of habitat

We used Area of Habitats (AOHs) for each species to represent and update their distribution range. We follow the methods detailed by Huang et al. (2021). AOHs indicate a species’ distribution by including published ranges (Brooks et al., 2019) and incorporating crowd-sourced data to identify the habitat and elevation at which species live. We acquired ranges from Birdlife for extant resident and breeding species (BirdLife International and Handbook of the Birds of the World, 2020) and overlapped them with the Northern Andes shapefile that Griffith, Omernik & Azevedo (1998) provide.

We accessed crowd-sourced data from eBird (eBird Basic Dataset, 2021) and GBIF (GBIF, 2022). For eBird, we used observations in checklists with less than 3 h in duration and a 2 km distance. These narrow limits reflect the topography of the Northern Andes and how elevation changes across short distances. For GBIF records, we use a three-step filtering process. First, we removed observations with “required” or “unverified coordinates”. We also removed observations with dubious remarks (under georeferenced remarks with keywords such as “no corresponde” or “no coincide”) or “no date”. Second, we built alpha hulls from the union of eBird presences and Birdlife ranges’ vertices. Alpha hulls derive from minimum convex polygons that exclude undetected areas and reduce bias in species range estimates (Burgman & Fox, 2003; Meyer, Diniz-Filho & Lohmann, 2017; Pateiro-López & Rodríguez-Casal, 2010). We set the required α-parameter (the radius distance below which all vertices of a Delaunay triangulation are retained and considered as observed areas) as the median inter-presence distance for each species. At this point, we selected all GBIF records occurring within the alpha hull. For records outside, we removed occurrences with evident bias related to distance (single isolated observations further than 50 km from the alpha hull) and source (collectors with systematic inconsistencies). We verified incorrect localities (non-forest habitats). Third, we generated a second and final alpha hull by incorporating the filtered GBIF records into the second step. For each set of data (eBird and GBIF), we removed duplicate coordinates.

We estimated each species’ elevational and habitat ranges within this final alpha hull, as Huang et al. (2021) suggest. For elevational ranges, we extracted values from a 90 m Digital Elevation Model (DEM) (Jarvis et al., 2008) based on each occurrence’s location. We removed the lower and upper 1% observations from the elevational distribution to reduce the influence of outliers. This threshold of 1% allows us to retain rare but accurate observations at extreme elevations but rejects unlikely extremes. We cropped the DEM within the alpha hull to the resulting elevation extremes. In cases where the literature mentioned more extreme elevations, we used them instead. As a habitat map, we used a 90 m tree cover layer from 2000 (Hansen et al., 2013). Within this layer, each cell represents a percentage of tree canopy coverage. We estimated a forest cover threshold from the upper 75% of the occurrence tree cover distribution. To some extent, all of our species depend on forests. We did not consider Páramo or Puna species, but we did include shrubland species that also rely on forest. The intersection of the elevation and tree cover layers within the alpha hull creates the AOH.

A few of our species lack records. For species without eBird observations, we applied the protocol for filtering GBIF occurrences. For species with no eBird nor GBIF records, we used the Birdlife range as the classic approach to estimate AOH (Brooks et al., 2019). We used the published elevational limits to trim the altitudinal ranges in such a case and did not set a threshold for tree cover but used all percentages instead.

We accumulated each AOH for all 335 bird species, so each 90 m raster cell represents several species. We refer to this stacked map as cumulative Areas of Habitat (cumulative AOH). This method allows us to determine priority geospatial areas based on species richness. The following steps describe conservation gaps and priority species methods according to their protection.

Estimating cumulative AOH within protected areas and community lands

We overlapped the cumulative AOH with state and private protected areas to identify conservation gaps. We accessed information from the Latin America and Caribbean region World Database of Protected Areas, WDPA (UNEP-WCMC, 2022) and complemented it with databases from each country’s protected areas repositories (Agencia Nacional De Tierras, 2017; Instituto del Bien Común, IBC)(2022; Ministerio del Ambiente Agua y Transición Ecológica Gobierno del Ecuador, 2022; Parques Nacionales Naturales De Colombia, 2022; SERNANP, 2022; SINAC-Costa Rica, 2021). We removed all protected areas within the marine environment and all internationally designated areas, such as Ramsar or World Heritage reserves, preserving coastal and terrestrial reserves. We then organised protected areas by Type of Governance and separated community lands for further analysis. We employed two of the five Types of Governance listed in the WDPA (government, shared governance, private, indigenous people, and Afro-Latin American communities). First, we merged state and shared governance types that refer to totally or partially managed by each country’s government and defined them as strictly state-managed areas. The second type of governance, private, is where the whole area is managed solely and strictly by private organisations.  For overlapping areas, we either trimmed them when they had different names, prioritising private over state areas or merged them if assuming name redundancy. For the community lands information, we accessed acknowledged indigenous territories and communities’ data from LandMark (LandMark, 2017) and did the same process with overlapping areas as with protected areas.

With the final set of protected areas across the entire cumulative AOH region, we estimated the relative proportion of protected cumulative AOH as the sum of species’ AOH within protected areas over the sum of all AOHs (units are in square kilometres). This metric gives a simple and direct figure of how protected species are without overestimations. The proportion of protection provides the overall percentage of cumulative AOH based on each AOH protection.

We proceeded the same way with indigenous lands. Indigenous and Afro-Latin American community-based reserves are culturally diverse and variably managed. Since they are not intrinsically protected areas, we did not include them in the prioritisation analysis but mapped their distribution to highlight their potential to fill spatial conservation gaps. We used Python 3.9 (Python Software Foundation, 2021), R 4.1.3 (R Core Team, 2020), and ArcGIS® Pro™ 3.0.2 by Esri (Redlands, WA, USA) to set, filter, and arrange databases and geospatial data.

Setting conservation priorities

Within the pool of all 335 range-restricted bird species, we prioritised those with limited protection and reduced AOH when compared to the area of published ranges. First, we plotted the percentage of protection against AOH size for every species, in which case we expected a trend. Globally, species with more extensive distribution ranges are proportionally less protected than species with smaller ones (Pimm, Jenkins & Li, 2018). We prioritised those species with small and relatively poorly protected AOH within our plot. From this subset of species, we mapped their cumulative AOH and identified the regions where they concentrate. These regions represent more targeted priorities than all species’ cumulative AOH. We list all subset conservation priorities indicating the AOH, protection, and IUCN categories. We map some species AOHs from this priority list with protected areas and community lands to understand the magnitude of some gaps and the relative contribution of these areas to their protection.

Determining protection along the elevational gradient

Finally, we looked at the extent of protection along the altitudinal gradient. We plotted the maximum species richness per elevational unit and estimated the relative number of protected areas from sea level to around 4,000 m.

Our analyses do not include threat layers such as proximity to roads, mining, agriculture, or even deforestation since our goal was to identify, at a regional level, conservation priorities based on range size and protection. At a local level, however, it would be essential to identify major threats to each priority.

Results

Cumulative AOH distribution

We analysed the current distribution and protection of 335 range-restricted and forest-dependent bird species across the Northern Andes. Their ranges stretch from the Amazon foothills in the east to the cloud forest in the inter-Andean valleys and the humid montane forest and lowland rainforest across the Darien gap north to Costa Rica. Bird species’ AOHs vary in size. Some species survive in less than 10 km2, while the largest species’ AOH is slightly smaller than 150,000 km2.

Aggregating 335 bird AOHs produced a cumulative Area of Habitat (cumulative AOH) that extends predominantly over six countries: Costa Rica, Panamá, Venezuela, Colombia, Ecuador, and Perú. This cumulative AOH does not reflect the total richness of countries beyond the Northern Andes limits (i.e., Peru, Costa Rica) but represents areas important to at least one small-ranged species. The maximum species richness is 58 and occurs in the Risaralda department north of the Tatamá National Park in Colombia.

We identified four conservation priority hotspots based on accumulated species richness (Fig. 1): Region I, the Sierra Nevada de Santa Marta and Serranía de Perijá (Colombia, Venezuela); Region II, the Mérida Cordillera and the Venezuelan Coastal Range (Venezuela); Region III, Western and Central Cordilleras (Colombia), Western and Eastern Cordilleras (north of Ecuador), and the Darien Gap (Panamá); and Region IV, the Tumbes and Marañón areas (Ecuador, Perú). The visibly largest and richest region lies in the Western Cordillera, from the Antioquia department in Colombia to the Pichincha province in Ecuador, right next to the Chocó bioregion hotspot. North of the Eastern Andes in Colombia contains additional minor hotspots.

Figure 1 The cumulative Area of Habitat (cumulative AOH) of 335 range-restricted bird species covering 1.3 M km2 across the Northern Andes.

We identified four priority areas for conservation: Region I, Sierra Nevada de Santa Marta and Serranía de Perijá; Region II, the Mérida Cordillera and the Venezuelan Coastal Range; Region III, Darien Gap, Western and Central Andes; and Region IV, the Tumbes-Marañón corridor. Basemap provided by Grupo Ingeo and Esri 2022 (Redlands, WA, USA).

Cumulative AOH protection and gaps

We found 4,215 protected areas spread across the whole extent of the cumulative AOH in the Northern Andes. Most are under private management (3,216), and the remaining are partially or entirely managed by governments (999). The aggregate size of state-protected areas is larger (842,721 km2) than private ones (8,557 km2). We discovered that the entire cumulative AOH, regardless of species concentration, is poorly protected. Only 31% is covered by State or Private areas in a highly fragmented pattern, while 69% remains unprotected (Fig. 2A). Elevating community lands to the conservation status would add 19% of protection connecting habitats in areas such as the western Andes and the Amazonian foothills between Ecuador and Perú (Fig. 2B). However, despite this latter addition, montane habitats largely would remain patchily protected.

Figure 2 Cumulative Area of Habitat (cumulative AOH) inside (yellow-green scale) and outside (white-brown scale) protected areas-PAs (A) and protected areas plus community lands-PAs+CLs (B).

Each map details protection of the richest regions: Western Andes and Tumbes-Marañón. Basemaps provided by Grupo Ingeo and Esri 2022 (Redlands, WA, USA).

Conservation priorities

A first approach to setting conservation priorities focuses on species with habitat ranges smaller than their published ones. 62% (207) of the total species analysed had such reduced habitat ranges (Table S1). The International Union for the Conservation of Nature, IUCN considers 77% (160) are not imperilled (Least Concerned and Near Threatened) out of this set of reduced habitat range species. Of these, 65% (104 spp.) have reduced or unknown population trends (see IUCN categories and trends in Table S1). Endemicity is also vital for prioritisation. Out of the 335 species, 45% (150 spp.) are unique to one of five countries: Colombia (74 spp.), Ecuador (6 spp.), Panamá (4 spp.), Perú (23 spp.), and Venezuela (43 spp.) (see Table S1).

After flagging species conservation priorities based on AOH size, we combined size and protection level to narrow the number of priority species, given that funding to protect species is limited. Species with larger AOHs are consistently less protected than those with small AOHs (Fig. 3). Within this species pool, we identified 50 top conservation priorities based on low protection (less than the median) and a small AOH range (less than 10,000 km2). These priority species require similar protection regardless of AOH size (see Table 1).

Figure 3 Percentage of protection for each of the 335 bird species’ AOHs.

The regression line shows that most small AOHs are proportionally better protected. We define conservation priorities as those small-range species (<10,000 km2) with low protection (dots under the dashed line), resulting in 50 priority species (pink dots).

Table 1 Top 50 conservation priority species in the Northern Andes according to the percentage of protection and AOH size.

The species list is ranked small to large AOH. Horizontal bars in the “AOH” column are scaled by numerical order, while in “Protection”, they are from 0 to 100%.

	

A map of cumulative AOH of our 50 high-priority species shows their distribution in part of the Sierra Nevada de Santa Marta, the three cordilleras in Colombia, the west Andes flank between Colombia and Ecuador, south of Ecuador and Northwest of Perú as well as the central East flank in Perú. We emphasise the central portion of the Western and Central Andes and the southern part of the Eastern and Central Andes in Colombia, the west Andes flank between Ecuador and Colombia, and the Marañón region between Ecuador and Perú as the areas with the highest concentration of these species (Fig. 4).

Figure 4 Cumulative Area of Habitat for 50 priority bird species and their spatial relation with protected areas.

These species have AOHs of less than 10,000 km2 and less than 34% (the median of protection for all species) of their habitat protected. Most species concentrate in Region IV of Fig. 1: the Seasonally Dry Tropical Forests of the Marañón. Basemap provided by Grupo Ingeo and Esri 2022 (Redlands, WA, USA).

Again, IUCN categories for all 50 species do not fully conform with their limited ranges and protection. Some 27species are not categorised as Threatened. Species such as the Chinchipe Spinetail (Synallaxis chinchipensis) and the White-masked Antbird (Pithys castaneus), classified as Least Concern (LC) and Near Threatened (NT), respectively, have no protection at all (Figs. 5A, 5B). Notably, the latter survives in a significantly reduced area (141 km2) (Table 1). Most of these species, treated as not threatened by IUCN, concentrate south of the Northern Andes between Ecuador and Perú, from the Marañón region to the central Peruvian Andes, where protected and community lands are sparse (see insets in Figs. 2A, 2B).

Figure 5 Distribution of some of the most endangered birds and their spatial relation with protected areas.

Some of the most critical species of our top priorities are (A) the Chinchipe Spinetail (Synallaxis chinchipensis), (B) the White-masked Antbird (Pithys castaneus), and (C) the Maranon Antshrike (Thamnophilus shumbae) with less than 1% protection. These birds share space along with other range-restricted and poorly protected species that make the Marañón region a high priority for conservation. Indigenous territories might offer some additional protection but governments must ensure protection across protection gaps. Alpha hulls around AOHs are for display purposes. Species images in order of appearance were kindly provided by Scott Terry, Raphaël Jordan, and Jon Irvine. Basemaps provided by Grupo Ingeo and Esri 2022 (Redlands, WA, USA).

On the other hand, some high-risk species are already recognised as Threatened. For example, the species with the smallest AOH (8 km2), the Gorgeted Puffleg (Eriocnemis isabellae), with three-quarters of its distribution currently unprotected, is Critically Endangered (CR) (Table 1). Similarly, the Maranon Antshrike (Thamnophilus shumbae), a vulnerable (VU) species with a bit over 1,000 km2 of remaining habitat, has merely 1% of protection (Table 1, Fig. 5C).

Protection along the elevational gradient

Elevational ranges for the 335 species vary. Some are restricted to a hundred meters above sea level, while others have been observed at almost 5,000 m (Table S1). Most accumulate at middle elevations, from roughly 1,000 m to 2,000 m (Fig. 6A). We were also interested in understanding protection along the altitudinal range of the cumulative AOH. Our data show increasing but not constant protection from the bottom to the top of the Northern Andes (Fig. 6B). We found a sizeable, unprotected portion in the lowlands compared to the whole area per elevation unit. Moreover, most areas above nearly 1,000 m of elevation have between 30% and 40% of uniform protection. Above around 3,500 m, protection starts to rise (Fig. 6B).

Figure 6 The cumulative Area of Habitat along the elevational gradient in the Northern Andes and its relative protection.

(A) Maximum species richness is high between 1,000 and 2,000 m and similarly low at extreme elevations. It is at these ranges that (B) protection maintains evenly between 30–40%. Relative protection is estimated as the amount of protected area divided by the total area and expressed in percentages.

Discussion

Conservation hotspots across the Northern Andes

Habitat loss and climate change challenge the high and unique diversity of range-restricted birds in the Northern Andes. We define conservation priorities by mapping the cumulative distribution of their remaining habitat in unprecedented detail. The four diversity hotspots we identify here are high-resolution representations of some of the region’s Endemic Bird Areas, for which Stattersfield (1998) describes two main threat categories. In Venezuela, the Mérida Cordillera is considered “critical”, while the Venezuelan Coastal Range is under an “urgent” category. In Colombia and Ecuador, all Endemic Bird Areas that totally or partially match our priority regions—the Chocó (Western Andes), the Northern Central Andes, the East Andes, the Santa Marta Mountains, and the Tumbesian region Endemic Bird Areas—are classed as “critical”. Finally, in Perú, the Marañón Valley and the Northeast Cordilleras are considered “urgent”. Here, we do not have tools to validate the threat of these regions, whose criteria are outdated (Rondinini et al., 2014). We consider all four regions critically important based on their high concentration of small-ranged species.

Our results confirm and expand previous studies that point to the Western Andes as the region with the most overlapping range-restricted species in the world (Fjeldså& Irestedt, 2009; Herzog & Kattan, 2012; Ocampo-Peñuela & Pimm, 2014; Pimm et al., 2014). The maximum number of forest-dependent birds with breeding ranges of <50,000 km2 within the western Andes is close to the maximum number of accumulated ranges estimated for the Chocó EBA (62 species, including non-forest species) (Long, Crosby & Stattersfield, 1996). Connecting and safeguarding the unprotected high-richness areas must be a priority.

Species with reduced range and low protection need critical attention

Relying on the IUCN red list categories to define at-risk species is misleading. Over three-quarters of species with reduced ranges would be mistakenly disregarded as priorities for conservation. IUCN does not categorise them as Threatened. We consider that the estimation of reduced range size (a primary reason for vulnerability) compared to the distribution is enough evidence to reconsider their conservation status. Previous studies have indicated the disparity of not threatened (Least Concern and Near Threatened) small-ranged species even though their ranges are similar or smaller than those judged as Threatened (Ocampo-Peñuela et al., 2016; Ocampo-Peñuela & Pimm, 2014). We understand the intricate process species must undergo to be classified as Threatened, but we also recognise the quick pace of habitat degradation. Therefore, we suggest considering species with reduced ranges as urgent conservation priorities.

Our more critical set of species are those with less than the median of protection for the entire set of species. They all should be considered Threatened due to their small habitat range (<10,000 km2), even though the IUCN categories do not reflect this currently, leaving over half the top species out of concern. Lack of protection adds a delicate layer of vulnerability to these species. For example, species without protection are listed as Near Threatened (NT) or of Least Concern (LC). They concentrate in areas of minimal and scattered protection across the Ecuador-Perú East Andes and the Marañón Valley EBAs near the Podocarpus and Colambo-Yacuri National Parks in Ecuador and the Tabaconas-Namballe National Park and the Cordillera del Collán in Perú. We must quickly connect and protect the gaps between these parks to reduce pressure on these high-risk species populations.

Collaborative management to address critical protection gaps and species

Current protected areas, assumed to work as modern refuges, poorly cover the four most important regions within the cumulative AOH extension. The two least protected are the two most critical areas, the Western Andes (Region III, Fig. 1) and the Tumbes-Marañón corridor (Region IV, Fig. 1). Likewise, the richest elevational band (1,000–2,000 m) has a low level of protection, a pattern consistent with South America as a whole (Elsen, Monahan & Merenlender, 2018). Given that most range-restricted birds are limited to middle elevational gradients and due to the increasing projected threats from climate disruptions in the inter-Andean valleys (Poveda et al., 2020) and at middle elevations overall in the Northern Andes (Medina, Huang & Pimm, 2023), we emphasise the importance of increased protection and connectivity of these elevational zones and of higher elevations where species may shift to in the future.

While birds overall and those with narrow ranges are well represented within protected areas in the tropics (Cazalis et al., 2020), their habitat ranges need to be more adequately represented. Increasing connectivity is critical (Kleemann et al., 2022; Marcelo-Peña et al., 2015; Ocampo-Peñuela & Pimm, 2014), and the responsibility primarily relies on nations with the most range-restricted species (those with small ranges and endemics). Interestingly, high levels of bird endemism strongly and positively correlate with pre-Columbian population centres in contrast to current population patterns (Fjeldså& Irestedt, 2009). Therefore, primary attention should be given to the Afro-Latin American territories in Colombia and Ecuador (the Chocó bioregion) and indigenous territories between Ecuador and Perú to help develop a just and collaborative immediate response within these areas. For example, the distribution of the zero-protection species mentioned above variably intersects community lands, which might offer some protection (Dawson et al., 2021; Estrada et al., 2022; Tran, Ban & Bhattacharyya, 2020).

However, community lands are not always where more threats occur. In Colombia’s inter-Andean slopes and valleys, heavily used by humans in the past and present, protection gaps are enormous but lack indigenous lands. Here, and in similar large, unprotected middle elevation areas across the Northern Andes, the responsibility lies on state and private reserves. Special attention must be given to private reserves, increasing their coverage (Bingham et al., 2021; Ocampo-Peñuela & Pimm, 2014; Pegas & Castley, 2016) and adopting similar management approaches for conservation-focused sustainable activities.

Conclusions

The quick pace of species loss worldwide requires conservationists to move beyond outdated threat measures and rapidly assess status based on the current ranges. After all, range reduction is one of the most critical risk assessment criteria. Our approach identifies conservation gaps and priorities based on the cumulative extent of the Area of Habitats for 335 range-restricted birds concentrated in the Northern Andes. We expect increasing management efforts to be allocated to the range extent of all 335 birds or at least to the 207 species for which ranges are smaller than previously thought. Realistically speaking, resources for conservation are scarce. Here, we suggest investing conservation efforts in 50 priority species and their cumulative distribution. Further assessments, including the effects of threats on ranges and populations, will help determine which of the four range-restricted bird hotspots across the Northern Andes are most critical.

How can land managers and stakeholders apply these suggestions to practical endeavours? First, identify protection gaps within the four priority conservation hotspots. Protection gaps encompass nearly 70% of the cumulative habitat of range-restricted birds in the Northern Andes and are more evident in the richest areas. Second, develop strategies to bridge these gaps. Strictly protected areas are more extensive, but their management is less flexible than private conservation areas. Private reserves and community lands can be equally effective in protecting biodiversity by following ground-up hierarchical organisation, securing people’s livelihoods, and conferring cultural, spiritual or economic value to their protective surroundings (Berkes, 2007). Governments must be accountable for supporting the creation of new protected areas that follow private and community-based practices within large-scale settings if they are to avoid the loss of the most vulnerable species while ensuring access to natural resources by local communities. Third, restore land to accelerate the recovery of degraded habitats. Priority species in Table 1 may be indicators for monitoring ecological restoration processes.

Organisations such as Saving Nature (http://www.savingnature.org) that implement such actions benefit from using priority species lists and credible, high-resolution maps to assess particular projects. Our experience is that other organisations also seek this advice. We anticipate that our methods and results may advise conservation priority settings of optimal areas to reforest and reconnect fragmented landscapes across the Northern Andes.

Supplemental Information

Supplemental Information 1 List of 335 forest-dependent range-restricted bird species in the Northern Andes

This dataset follows Clements’s (2020) taxonomical order and provides scientific names used by both Birdlife and eBird.

We appreciate the valuable comments and suggestions of two anonymous reviewers who helped improve early versions of this manuscript. This study contributes to a bioeconomic conservation focus to sustain biodiversity under the Colombia Científica program.

Additional Information and Declarations

Competing Interests

Author Contributions

Data Availability

Stuart L. Pimm is an Academic Editor for PeerJ but he does not take part in the selection of nor influence the suggested editors. The other authors declare that they have no competing interests.

Wilderson Medina conceived and designed the experiments, performed the experiments, analyzed the data, prepared figures and/or tables, authored or reviewed drafts of the article, and approved the final draft.

Stuart L. Pimm conceived and designed the experiments, analyzed the data, authored or reviewed drafts of the article, and approved the final draft.

Ryan M. Huang conceived and designed the experiments, analyzed the data, authored or reviewed drafts of the article, and approved the final draft.

The following information was supplied regarding data availability:

The raw data are available in the Supplemental File.

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
