# Peer review of "Conservation gaps and priorities of range-restricted birds in the Northern Andes"

_PeerJ, doi:10.7717/peerj.16893_

## Round 0.1 · original submission · Major Revisions

Although the reviewers recognized merits, they mention limitations and drawbacks, raising some misgivings about the way the manuscript has been written and analyzed. They pointed out that a clear rationale explaining why the study area was focused only in the Northern Andes is needed, the methods should be revised, and a more in-depth discussion including practical conservation perspective is recommended. I hope that you will find all advice helpful when revising the manuscript.

Reviewer 1 ·

Basic reporting

This study aims to: devise high-resolution range maps for the range-restricted bird species of the northern Andes, identify hotspots of those species, and determine the percentage of species’ ranges that are currently protected. Range maps are represented by AOH (Area of Habitat), which is calculated based on the combination of citizen science data (eBird and GBIF), elevational ranges, and habitat associations. AOHs are calculated for all species and compared to the BirdLife (BL) ranges. A composite of species’ AOHs is used to identify hotspots, resulting in four foci of diversity. Around a third of species’ ranges fall within protected areas.

I could not agree more that accurate range maps are critical for conservation of these range restricted species. I think this study clearly identifies that need, and the methods seem mostly appropriate. Certainly, the calculations involving protected areas are straightforward. I like the figures and table. However, I do have some questions/concerns about the data on which the AOH is based, and the AOHs that come out from this process.

Further comments:
L93: Why restrict this study to the northern Andes? How is that region defined? Could the same process not be applied to the entire tropical Andes? Certainly, central and southern Peru has more than its fair share of range-restricted species.

L268: Is “ponder” the right word here?

L285: I think it is also really important to emphasize that, because species are shifting upslope, protected areas should also consider where species are likely to be in the future!

L292-298: Agreed!

Figure 2: How well does the green and magenta do for red-green color blindness?

Figure 4: Perhaps indicate in the caption where 32% comes from as a number.

Table 1: I really like the filled bars in this table.

Table S1: It would be useful to have a column that orders species taxonomically.

Experimental design

It seems problematic to include checklists up to 7 km long. This distance is a huge distance and could span a non-trivial elevational gradient. For example, the famous Km 18 hotspot near Cali, Colombia, is around 2000 m in elevation. Driving 7 km back down the road towards Cali takes you to 1600 m. This change in elevation is not nothing, especially for something like a cloud forest tapaculo, as you move from cloud forest to foothill forest. Furthermore, the best practices, while useful, are been based mostly on temperate species, are not universally applicable, and ought to be tailored to the question being asked. I strongly suggest that this study considers the effect that such long distances could have on the results. Perhaps re-run the AOHs with more stringent filtering?

Further comments:
L128: Is 1% stringent enough? Because of the long distance of checklists, species from higher (or lower) elevations could be assigned to checklists at inappropriate elevations.

L163-165: What about calculating the average of the percent of protection for each species?

A species I found missing from the list was the recently described Blue-throated Hillstar (Oreotrochilus cyanolaemus). Is there a reason why it is missing? It has both a BirdLife range and 215 eBird records. Are other species absent from the list?

Validity of the findings

I also seriously question some of the AOH results to the point that it seems like there must be a big issue in how they are calculated. In inspecting Table S1, I sorted based on AOH. The top species is the endangered Spinus cucullatus (Red Siskin) with an AOH of almost 220,000 km2. That is compared to a BL area of 3250 km2, two orders of magnitude less. Red Siskin only has 178 records on eBird, most of which fall outside of the northern Andes in Guyana. There is no way that this species has a range of 220,000 km2 but only 178 eBird records. Something wrong must be going on here? The next case down on the list is Cypseloides Lemosi; similar situation: AOH of 203,000 km2, BL area of 3580 km2, 626 eBird records. Maybe these are a couple of outliers in an otherwise good process, but it gives me serious pause when considering the accuracy of the AOHs in general.

Further comments:
L192: Upper limits in Table S1 max at 5,107 m?

L199: Half is an interesting proportion to reach. It suggests that ranges are either too large or two small roughly 50% of the time, but it’s is hard to know, then, what the gold standard is. For example, if AOHs were consistently smaller than BL areas, we could say that BL is overestimating ranges. But the 50:50 makes it seems rather random. Can we draw any inference from this?

L224: It says 22% in the abstract, or is that a different number?

Reviewer 2 ·

Basic reporting

The manuscript titled "Conservation Gaps and Priorities of Range-Restricted Birds in the Northern Andes" addresses an important topic concerning the conservation of range-restricted bird species in a highly biodiverse region. This manuscript makes a valuable contribution to the field of avian conservation by focusing on range-restricted bird species in the Northern Andes, which are often overlooked in broader conservation discussions. However, the authors should include a layer that includes the anthropogenic pressure in the studied area, not only the geospatial terms. In addition, perform including an analyses on the functional diversity of the species, and not just the richness, to identify the loss of certain functions that the species exert in this region of the Andes.
Specifically, at line 23, revise the number of species (you cite 355 species). I suggest that you improve the description at line 193 into Results section, to provide more details and references for the affirmation that middle elevation includes a large range of average elevation of tree line from 500 to 3,000m. The authors should explain the reasons to not include a threaten layer on the analysis to define de priorities areas. The size of the area of habitat and the occupied area are the result of the pressure exerted by the threats. I suggest to include in you discussion this subject.
I recommend the authors delve deeper into the specific conservation gaps and priorities identified for the 45 conservation priority bird species. Providing more detailed information and examples for the 45 conservation priority species would enhance the manuscript's practical applicability and assist conservation practitioners in prioritizing their efforts effectively.
Furthermore, it would be beneficial to include a section discussing the potential conservation actions and strategies that could address the identified gaps and priorities. Elaborating on the proposed measures, such as habitat protection, restoration initiatives, community engagement, and international collaborations, would enhance the manuscript's relevance and applicability for conservation practitioners and policymakers.
Overall, this manuscript has the potential to significantly contribute to the field of avian conservation, specifically focusing on range-restricted bird species in a critical region. With the suggested revisions, I believe this work will make a valuable addition to the scientific literature and should be considered for publication in PeerJ.

Experimental design

No comment.

Validity of the findings

No comments.

---

## Round 0.2 · Minor Revisions

Dear authors,

Thank you very much for sending a revised version of your manuscript. However, there are some points that need to be solved/clarified before a final decision is made. I hope that you will find all the advice helpful when revising the manuscript.

Reviewer 1 ·

Basic reporting

The authors did a great job revising the manuscript. I particularly appreciated the more stringent filters, and it is good to see that didn’t change the overall results. I only have a few minor comments:

L15: Should “Tropics” be a proper noun?

L61-91: The latter half of the intro gets a bit choppy, with some very short paragraphs, jumping a bit between backstory and aims. Perhaps streamline a bit.

L150: Consider subheadings for the methods.

Figure 1: I still find it odd that southern Peru is shown here despite not being in the Northern Andes. I understand that restricted species from the Northern Andes extend that far south, but it creates the (misleading) impression that southern Peru lacks range-restricted species, which, of course, it doesn’t, they just weren’t considered in this study.
Could you also lighten the grey of the map so that the colours pop more?

L245: Is it a coincidence that it’s 50? It seems like you picked a top 50, rather than the number of species fitting those criteria being exactly 50. Maybe clarify.

L270: I think “Elevational” would technically be more correct than “Altitudinal”.


L331: “evenly low”? Unsure what that means.

L353-355: The grammar in this sentence was a bit off.

Conclusion: I think this section could be punchier. The first sentence lacks context (imagine if that was all the reader read). The last sentence about space limits is also a very odd way to finish.

Experimental design

NA

Validity of the findings

NA

Reviewer 2 ·

Basic reporting

No comment

Experimental design

No commet.

Validity of the findings

No comment.

---

## Round 0.3 · accepted · Accept

After a careful reading of the revised version of this manuscript, I would like to express my appreciation to the authors for their good job in answering all the questions raised and I am happy to accept this manuscript in its current form.